# Effects of heat waves on cardiovascular and respiratory mortality in Rio de Janeiro, Brazil

**Ismael H. Silveira**[1]*, **Taísa Rodrigues Cortes**[2], **Michelle L. Bell**[3], **Washington Leite Junger**[2]

**1** Institute of Collective Health, Federal University of Bahia, Salvador, Brazil, **2** Institute of Social Medicine, Rio de Janeiro State University, Rio de Janeiro, Brazil, **3** School of the Environment, Yale University, New Haven, Connecticut, United States of America

* ismaelhsilveira@gmail.com

## Abstract

### Background

Heat waves are becoming more intense and extreme as a consequence of global warming. Epidemiological evidence reveals the health impacts of heat waves in mortality and morbidity outcomes, however, few studies have been conducted in tropical regions, which are characterized by high population density, low income and low health resources, and susceptible to the impacts of extreme heat on health. The aim of this paper is to estimate the effects of heat waves on cardiovascular and respiratory mortality in the city of Rio de Janeiro, Brazil, according to sex, age, and heat wave intensity.

### Methods

We carried out a time-stratified case-crossover study stratified by sex, age (0–64 and 65 or above), and by sex for the older group. Our analyses were restricted to the hot season. We included 42,926 participants, 29,442 of whom died from cardiovascular and 13,484 from respiratory disease, between 2012 and 2017. The death data were obtained from Rio de Janeiro's Municipal Health Department. We estimated individual-level exposure using the inverse distance weighted (IDW) method, with temperature and humidity data from 13 and 12 stations, respectively. We used five definitions of heat waves, based on temperature thresholds (90th, 92.5th, 95th, 97.5th, and 99th of individual daily mean temperature in the hot season over the study period) and a duration of two or more days. Conditional logistic regression combined with distributed lag non-linear models (DLNM) were used to estimate the short-term and delayed effects of heat waves on mortality over a lag period (5 days for cardiovascular and 10 for respiratory mortality). The models were controlled for daily mean absolute humidity and public holidays.

### Results

The odds ratios (OR) increase as heat waves intensify, although some effect estimates are not statistically significant at 95% level when we applied the most stringent heat wave criteria. Although not statistically different, our central estimates suggest that the effects were

**Data Availability Statement:** Data availability statement The mortality data used in this study are protected by the General Personal Data Protection Law (No. 13.709) and the National Health Council Resolution (No. 466/12) and cannot be made

available due to potential identifying information (such as the residential address). Access to the mortality data can only be obtained after submitting a complete research project to the Research Ethics Committees at https://plataformabrasil.saude.gov.br/. After approval by the Research Ethics Committees, the mortality data can be requested from the Municipal Health Department of Rio de Janeiro (https://www.rio.rj.gov.br/web/sms/vigilancia-em-saude). Meteorological data can be obtained from the websites of the respective organizations responsible for providing the data, described below: • Data from DCEA (Departament of Airspace Control) were obtained from the PROTIM/CPTEC/INPE (Portal de Tecnologia da Informação para Meteorologia, Centro de Previsão do Tempo e Estudos Climáticos, Instituto Nacional de Pesquisas Espaciais) system, website currently not working. Nowadays, the dataset from DCEA can be downloaded at: https://bndmet.decea.mil.br/. • Data from INMET (National Institute of Meteorology) can be obtained from the INMET meteorological database system (BDMEP) at https://bdmep.inmet.gov.br/#; • Data from SMAC (Rio de Janeiro Municipal Secretariat for the Environment) can be obtained from the website DATA.RIO (https://www.data.rio/maps/PCRJ::qualidade-do-ar-dados-hor%C3%A1rios/about).

**Funding:** This research was funded by the Coordination for the Improvement of Higher Education Personnel – CAPES (https://www.gov.br/capes/) (finance code 001), the Foundation for Research Support of the State of Rio de Janeiro-FAPERJ (https://www.faperj.br/) (grant numbers E-26/010.002131/2019 e E-26/200.966/2022), the National Council of Technological and Scientific Development – CNPq (https://www.gov.br/cnpq/) (grant numbers 307495/2018-3, 406292/2018-3 and 315349/2021-2) and the Wellcome Trust (https://wellcome.org/) (grant number 216087/Z/19/Z). The funders had no role in study design, data collection and analysis, decision to publish, or the preparation of the manuscript.

**Competing interests:** The authors have declared that no competing interests exist.

greater for respiratory than cardiovascular mortality. Results stratified by sex and age were also not statistically different, but suggest that older people and women were more vulnerable to the effects of heat waves, although for some heat wave definitions, the OR for respiratory mortality were higher among the younger group. The results also indicate that older women are the most vulnerable to heat wave-related cardiovascular mortality.

## Conclusion

Our results show an increase in the risk of cardiovascular and respiratory mortality on heat wave days compared to non-heat wave ones. These effects increase with heat wave intensity, and evidence suggests that they were greater for respiratory mortality than cardiovascular mortality. Furthermore, the results also suggest that women and the elderly constitute the groups most vulnerable to heat waves.

## Introduction

Heat waves, meteorological events characterized by unusually high temperatures sustained for prolonged periods, can affect human health and survival, and are among the most dangerous environmental hazards [1]. Epidemiological evidence reveals the health impacts of heat waves on numerous mortality and morbidity outcomes, including hospital admissions, emergency department visits, ambulance attendances, occupational injuries, all-cause and cause-specific mortality [2–6]. The biological mechanisms that underlie the effects of heat waves on human health are not fully understood, although many physiological responses to extreme temperatures may be related to cardiorespiratory events [2]. Cardiorespiratory diseases are the leading cause of morbidity and death worldwide [7], and numerous epidemiological studies suggest an increase in cardiorespiratory events during heat waves, although the magnitude of these effects varies across countries and population groups [2].

The global mean temperature has been increasing since the pre-industrial era, and extreme heat events are becoming more intense and frequent [8]. As a consequence of climate change, the temperature will continue to rise, and in the future heat waves will be even more frequent and severe [9]. According to the latest report from The Lancet Countdown, vulnerability to the extremes of heat and exposure to heat waves have risen worldwide [10]. In the context of both climate change and ageing populations, the impacts of heat waves on health are expected to intensify [10, 11].

Although the health effects of heat waves are well documented, most study locations are concentrated in high income countries, principally in Europe, North America and Australia; few studies have been conducted elsewhere, except in China [3, 12]. There is a scarcity of studies in tropical regions characterized by high population density, low income and low health resources, which are susceptible to the impacts of extreme heat on health [3]. Investigations to improve the description of the health risks arising from heat waves in these regions are urgently required, to promote public policies to mitigate and adapt to the impacts of the climate crisis.

To the best of our knowledge, few epidemiological studies of heat waves-related mortality have been conducted in Brazil [12, 13]. One multi-city study examined the impacts of heat waves on all-cause mortality and reported the results at national level [14]. Another study conducted in 32 municipalities in the Brazilian Amazon, examined the effects of various heat wave definitions according to sex, age group, and cause of death [15]. Two studies were conducted

in the city of São Paulo, estimating the effects of heat waves on all-cause [16] and elderly mortality [17], while two studies, comparing observed and expected deaths during heat waves, were performed for the Metropolitan Region of Rio de Janeiro [18, 19].

The aim of this paper is to estimate the effects of heat waves on cardiovascular and respiratory mortality in the city of Rio de Janeiro, Brazil between 2012 and 2017. We stratified our analysis by sex, age, and a combination of both strata, and assessed how these effects change according to heat wave intensity. Different to most previous study conducted in Brazil that used a single city exposure measure for all participants, we used an estimate of individual-level exposure, in order to account for the spatial variability of weather conditions and reduce exposure misclassification error.

## Methods

### Mortality data

We obtained data on deaths from cardiovascular and respiratory diseases (codes I00-I99 and J00-J99 of the 10th Revision of the International Classification of Diseases), from Rio de Janeiro's Municipal Health Department. We only included deaths among residents of the city of Rio de Janeiro that occurred during the hot season (November to March) of our study period (2012–2017). Residential addresses were geocoded using Google Maps, as described in our previous work [20]. In summary, of a total of 132,863 deaths, 113,876 (86%) were geocoded at street level, with 87% sensitivity and 98% specificity.

We obtained ethical approval for this study from the Research Ethics Committees of the Municipal Health Department of Rio de Janeiro and the State University of Rio de Janeiro. The mortality data were anonymized, and the authorization to carry out the research without the informed consent of subjects was given by the ethics committees.

### Weather data

Temperature and relative humidity data, from 2012 to 2017 (during the hot season), were obtained from the stations of Brazil's National Institute of Meteorology–INMET, from Rio de Janeiro's Municipal Secretariat for the Environment, and from airports run by the Brazilian Air Force's Department of Airspace Control. We only selected stations with less than 20% missing data per year, and less than 15% for the entire period. We therefore used 13 stations for temperature data and 12 for relative humidity. The station locations can be found in S1 Fig. Additional information about data availability can be found in the S1 Table.

In this study we used daily series for mean temperature and mean relative humidity by station as inputs for individual-level exposure estimation, detailed below. Some meteorological stations provided daily measures of temperature and humidity (including minimum, mean and maximum values), while others provided hourly values (S1 Table) for which the daily means were calculated by averaging these hourly values. Missing values were imputed using a method based on the expectation-maximization algorithm, which considers dependence between variables and the temporal dependence of each variable, implemented using the R mtsdi package [21]. We modeled temporal components, long-term trend and seasonality, using natural cubic splines with 5 degrees of freedom per year. Information about imputed missing data for each weather variable daily series is presented in the S1 Table.

### Individual-level exposure estimation

We estimated the individual-level exposure to temperature and humidity using the inverse distance weighted (IDW) method. The IDW method is used to estimate exposure values at

unmeasured locations by averaging the weighted sum of values from their nearest neighbors (monitoring stations) within a search radius. The weights are a function of the inverse distance between the measured and unmeasured locations [22].

We used leave-one-out cross-validation to determine the IDW parameters (the power value, number of nearest neighbors, and search radius), which minimized the prediction of root mean square error (RMSE) [23]. We evaluated the power of distance with five values from 1 to 3, varying the number of neighbors from 1 to the maximum number of stations (12 or 14). The root mean square error (RMSE) and mean absolute error (MAE) values of cross-validation are presented in the S2 Table.

We interpolated the daily weighted averages of temperature and relative humidity using 1 $km^2$ grids (Hengl, 2006). Exposures at individual level were estimated by averaging the interpolated values within a 1km radius buffer centered on each residential address. The interpolation was conducted in R, using the gstat package. We also calculated the daily mean individual level exposure to absolute humidity, based on temperature and relative humidity, using the HeatStress package for R.

## Heat wave definitions

There is no universal definition for a heat wave, although it is typically based on a temperature metric, intensity (an absolute or relative threshold) and duration [1, 4]. Here, we used five heat wave definitions, using five different temperature thresholds (90th, 92.5th, 95th, 97.5th, and 99th of the daily mean temperature in the hot season, estimated for each participant as described above, over the study period) and lasting two or more days. We chose to work with heat waves lasting at least two days, since this is a less restrictive definition, and evidence has suggested that duration plays a less important role than intensity in determining the magnitude effect of a heat wave [4]. The S3 Table shows the definitions of heat waves we used and their respective mean threshold values, based on individual threshold values.

## Study design and statistical analysis

We carried out a time-stratified case-crossover study to estimate the effect of heat waves on cardiovascular and respiratory mortality in the city of Rio de Janeiro, Brazil, between 2012–2017. We restricted our analysis to the hot season (November to March). We stratified our analysis by sex, age (0–64 years old, 65 years or above), and by sex for the older group (older men and women). Case-crossover is an epidemiological study design suitable for estimating the effects of short-term exposure to the risk of acute events, which uses a sample of subjects that experience the event of interest in order to compare exposure on the day of the event (index day) with exposure at different times (control times) [24, 25]. We used the time-stratified referent selection strategy by restricting controls to the same day of the week, month, and year as the death [26].

A causal directed acyclic graphic (DAG) was developed in order to state our study hypotheses and identify the sufficient adjustment set of variables in our models to estimate the total effect of heat waves on mortality (S2 Fig). In case-crossover studies, potential time-invariant confounders (e.g. age, sex, personal habits and behavior, etc.) are adjusted by design, because participants are compared with themselves [24]. The time stratified control selection approach also adjusts the potential confounding effects of day of the week, seasonality and long-term trend by design [26]. Thus, our sufficient set of variables to estimate the total effect of heat waves on mortality is composed of humidity and public holidays. We built our causal diagram in DAGitty [27].

Conditional logistic regression models combined with distributed lag non-linear models (DLNM) were used to estimate the short-term and delayed effects of heat waves on mortality. We used a dummy variable to identify heat wave days (1 for heat wave days, and 0 for non-heat wave days) and the DLNM structure was incorporated to account for the delayed effects of heat waves on mortality [28]. We built a cross-basis function with a linear function to represent the effect of heat waves, and a natural cubic spline with 4 degrees of freedom (2 equally spaced knots along their logarithmic scale) for the 5-day lag for cardiovascular, and the 10-day lag for respiratory mortality. In our study, 5 and 10 lag days seemed sufficient to represent the delayed effect on cardiovascular and respiratory mortality respectively, and account for potential mortality displacement (S3 Fig). The selection of the spline function and number of knots was based on a minimization of the sum of the Akaike Information Criteria for the models, taking account of the 5 heat wave definitions and the 2 mortality outcomes analyzed. A dummy variable was included to represent public holidays. We included a cross-basis function for absolute humidity, using a natural cubic spline with 4 degrees of freedom for the predictor, and a b-spline with 4 degrees of freedom for the lag period (5 or 10 days). Absolute humidity is recommended as a better metric for humidity than relative humidity, which may reduce over-fitting [29].

We performed some sensitivity analyses changing the parameters of the cross-basis function to represent heat waves (using a quadratic b-spline for lags, 3 and 5 degrees of freedom for the natural cubic splines), the number of lag days (3 and 10 for cardiovascular mortality, and 3 and 5 for respiratory mortality), and the adjustment sets of variables (models without absolute humidity and including mean ambient temperature). We adjusted for temperature with a cross-basis, using a natural cubic spline with 4 degrees of freedom for the exposure and the 5-day lag for cardiovascular, and 10-day lag for respiratory mortality. Our analyzes were conducted in R, using the survival and dlnm packages.

## Results

We recorded a total of 132,863 deaths from cardiorespiratory outcomes, which occurred in Rio de Janeiro between 2012 and 2017. We excluded deaths with non-geocoded addresses and those with lagged exposure in relation to the index or a referent period prior to 2012. We also restricted our analysis to the hot season. This resulted in 42,926 participants, 29,442 of whom died from cardiovascular and 13,484 from respiratory disease. Table 1 shows the summary statistics for the study population and the weather variables over this period.

Table 2 shows the odds ratios (OR), and their 95% confidence intervals for the cumulative effects of heat waves of varying intensity on cardiovascular and respiratory mortality, for the entire study population, and stratified by age and sex. In general, the effect estimates rise as the heat wave intensifies. However, in some strata, when we use the most stringent heat wave criteria, the effect estimates are not statistically significant at 95%. Although the results were not statistically different, in general we found greater effect estimates for respiratory than cardiovascular mortality, but for the most stringent definition, the central OR decreased and become very imprecise.

Results stratified by sex and age were not statistically different but suggest that the older people and the women were more vulnerable to the effects of heat waves, although for some heat wave definitions, the odds ratios were higher among men and the younger group. There was also an increase in the mortality odds ratio for men during heat waves, but this was lower than for women, and all values demonstrated uncertainty above 5%. When we stratified our analysis of the older group by sex, for cardiovascular mortality, the results suggest that older women are the most vulnerable to heat wave effects, with higher central estimates than older

**Table 1. Study population summary statistics and weather variables in Rio de Janeiro, Brazil, 2012–2017.** The data were restricted to the hot season (November to March).

| | N | % |
|---|---|---|
| **Cardiovascular mortality** | 29,442 | |
| Male | 14,047 | 47.7 |
| Female | 15,395 | 52.3 |
| 0–64 years | 8,085 | 27.5 |
| 65 or above | 21,347 | 72.5 |
| **Respiratory mortality** | 13,484 | |
| Male | 6,067 | 45,0 |
| Female | 7,417 | 55,0 |
| 0–64 years | 2,404 | 17,8 |
| 65 or above | 11,080 | 82,2 |
| **Weather variables** | | |
| Average temperature (˚C)[a] | 27.7 | 2.5 |
| Individual-level temperature (˚C)[b] | 27.8 | 2.7 |
| Average relative humidity (%)[a] | 70.9 | 9.7 |
| Individual-level relative humidity (%)[b] | 69.1 | 10.5 |

[a]Average values based on weather stations.

[b]Mean values for individual-level exposure estimation.

men, except during the most intense heat waves (using the 99th percentile). For respiratory disease, older women were more vulnerable during the less intense heat waves (using the 90th and 92.5th percentiles), while older men were more vulnerable to heat waves of greater intensity (using the 95th, 97.5th and 99th percentiles).

The results of our sensitivity analyses are shown in the S4 Table. We observed very similar results when we changed the cross-basis parameters for the heat wave adjustment (using a quadratic b-spline for lags, and changing the degrees of freedom of the natural cubic spline to 3 and 5). Although not statistically different, the odds ratios were slightly lower when using 3-

**Table 2. Odds ratios (OR) and 95% confidence intervals (95% CI) for the cumulative effect of heat waves on cardiovascular and respiratory mortality, over 5 and 10-day lags respectively, in the city of Rio de Janeiro, Brazil, stratified by age and sex.**

| HW threshold | OR (95% CI) | | | | | Older men | Older women |
|---|---|---|---|---|---|---|---|
| | All | Older (> = 65) | Younger (0–64) | Men | Women | | |
| **Cardiovascular mortality** | | | | | | | |
| 90th | 1.26 (1.14–1.40) | 1.27 (1.13–1.44) | 1.23 (1.01–1.50) | 1.11 (0.95–1.29) | 1.42 (1.23–1.64) | 1.07 (0.89–1.29) | 1.45 (1.23–1.70) |
| 92.5th | 1.31 (1.15–1.49) | 1.36 (1.17–1.59) | 1.16 (0.90–1.50) | 1.12 (0.93–1.35) | 1.51 (1.26–1.81) | 1.10 (0.87–1.39) | 1.60 (1.31–1.96) |
| 95th | 1.38 (1.15–1.66) | 1.46 (1.18–1.81) | 1.19 (0.84–1.70) | 1.29 (0.99–1.67) | 1.48 (1.15–1.91) | 1.20 (0.86–1.67) | 1.69 (1.28–2.24) |
| 97.5th | 1.26 (0.86–1.83) | 1.28 (0.83–2.00) | 1.19 (0.58–2.45) | 1.23 (0.71–2.10) | 1.30 (0.77–2.19) | 1.13 (0.57–2.23) | 1.41 (0.79–2.52) |
| 99th | 1.76 (0.80–3.88) | 1.53 (0.61–3.85) | 2.52 (0.54–11.79) | 1.95 (0.64–5.95) | 1.64 (0.54–5.00) | 1.75 (0.44–6.95) | 1.33 (0.38–4.64) |
| **Respiratory mortality** | | | | | | | |
| 90th | 1.36 (1.03–1.81) | 1.43 (1.05–1.95) | 1.06 (0.53–2.12) | 1.18 (0.78–1.79) | 1.54 (1.05–2.25) | 1.23 (0.76–1.97) | 1.61 (1.07–2.42) |
| 92.5th | 1.69 (1.19–2.40) | 1.62 (1.10–2.39) | 2.04 (0.88–4.75) | 1.51 (0.90–2.53) | 1.85 (1.14–2.99) | 1.50 (0.83–2.72) | 1.71 (1.02–2.86) |
| 95th | 2.00 (1.23–3.25) | 2.06 (1.20–3.52) | 1.64 (0.50–5.37) | 1.74 (0.85–3.56) | 2.22 (1.14–4.34) | 2.06 (0.90–4.70) | 2.05 (1.00–4.17) |
| 97.5th | 4.17 (1.55–11.18) | 5.08 (1.72–15.01) | 1.54 (0.14–17.08) | 2.73 (0.64–11.70) | 5.63 (1.47–21.64) | 5.90 (1.13–30.7) | 4.25 (1.00–17.97) |
| 99th | 1.15 (0.15–8.90) | 1.54 (0.16–14.5) | 0.32 (0.00–50.8) | 1.51 (0.08–27.34) | 0.66 (0.04–12.46) | 3.54 (0.13–95.12) | 0.65 (0.03–14.32) |

and 10- day lags for cardiovascular, and 3- and 5- day lags for respiratory mortality, and were slightly higher when absolute humidity was removed from the model. When we controlled for temperature, the added effect estimates of heat waves were observed for almost all heat wave definitions, except in the most intense heat waves (defined at the 99[th] percentile of daily mean temperature) for respiratory mortality, although most were not statistically significant at the 95% level.

## Discussion

In this study, we used a case-crossover design based on individual-level exposure estimates to investigate the effect of heat waves of varying intensity on cardiovascular and respiratory mortality, stratified by age and sex subgroups, in the city of Rio de Janeiro, Brazil. Our results show an increase in the risk of cardiovascular and respiratory mortality on heat wave days compared to non-heat wave ones, and suggest that the effects were greater for respiratory than cardiovascular mortality, consistent with the conclusion of a systematic review and meta-analysis [2]. In general, the effects increase as heat waves intensify, which is also in accordance with the literature [4]. The evidence suggests that women and elderly are more vulnerable to heat waves, as indicated by previous systematic reviews [12, 30]. Furthermore, when we stratified our analysis of the older group by sex, older women were more vulnerable to cardiovascular mortality, while the results for respiratory mortality were inconclusive, suggesting that, depending on heat wave intensity, both groups were vulnerable.

Our findings are in agreement with evidence from previous studies conducted in Brazil, most of which use single-city exposure measures and different study designs [16, 17, 31]. These studies observed that heat waves have a greater effect on respiratory than cardiovascular mortality, with greater risk from higher intensity heat waves. Son et al. [16] analyzed the modification of heat wave effects in the city of São Paulo on total non-external mortality and observed that estimates increase with age and are greatest for those aged 75 or above. Moraes et al. [31], in a study restricted to older people ($> = 65$ years) and conducted in the city of São Paulo, found that mortality risk during heat waves was greater for total cardiovascular and respiratory diseases among women. The study of Diniz [17], also restricted to people aged 65 or above and conducted in the Metropolitan Region of São Paulo, found that heat waves have a greater effect on women for both cardiovascular and respiratory outcomes. A study in the Metropolitan Region of Rio de Janeiro [18] observed high numbers of excess deaths during four heat wave events in 2010 and 2012, and the observed-expected death ratios were higher for women than men and for the older group. In a recent paper conducted in the Brazilian Amazon [15], we also found that heat wave effects increase with heat wave intensity and were greater for the elderly and women, however, as opposed to this study, there, the effects were greater for cardiovascular mortality than for total non-external and respiratory mortality.

In previous research conducted in Rio de Janeiro [32], we found that the cardiovascular mortality risk increased with ambient temperature, and estimates of the heat effect were greater for older people and women. In the present study, our sensitivity analyzes suggest that, beyond the temperature effect, an additional effect arises from the duration of heat sustained over several consecutive days, in line with Gasparrini and Armstrong [33].

The heterogeneity effect of heat waves across population subgroups can be explained by several factors. In general, older people may have other comorbidities, which can make it harder for the body to respond to thermal exposures. But even in healthy older people, the changes in cardiovascular function that occur with aging can compromise the body's ability to perform thermoregulation during exposure to heat waves [34]. Gender or sex differences can be explained by differences in age structure, occupational and socioeconomic status, mobility

and exposure patterns, in addition to physiological differences in coping with heat [12, 35]. Further, in our study, the mean age of women (82.4 years) in the older group is higher than that of men (78.9 years). Women may have greater difficulty in dissipating heat due to their fitness levels, the higher percentage of body fat, and possible differences in skin conductance. Furthermore, differences in hormone levels, especially the changes in post-menopausal estrogen levels, may be associated with inflammatory responses during thermoregulation [35].

We estimated residential temperature and humidity levels at individual resident addresses, based on the inverse distance weighting method, in order to account for the spatial variability of the weather factors, and reduce the exposure misclassification error. Nonetheless, this estimation method has limitations, since it is based on the values of the observed variable at the weather stations, rather than taking other determinants of the exposure variable into account, such as altitude, land cover, proximity to water, solar radiation, wind speed, etc. The Guo et al. [36] study of ordinary kriging, which took account of other environmental variables, provided better spatial predictions of temperature than inverse distance weighting. Although we used an individual-level exposure estimation, we did not consider other determinants of individual exposure related to daily mobility or an environment with warmer or milder temperatures, for example an air-conditioned one.

Another study limitation is the loss of deaths that could not be geocoded, which may have influenced our effect estimates. Losses during geocoding tend to be concentrated in places with poorer urban infrastructure, where cartographic databases tend to be of low quality, coinciding with areas of lower income and greater segregation, which are essential health determinants. There was a higher proportion of losses among Black and mixed race people and those with lower educational levels compared to the study population, 54% vs 37%, and 74% vs 61%, respectively. However, we believe that, if present, any such bias would have distorted our results towards the null. Further, the geocoding of residence does not account for differences in how travel and indoor/outdoor activity patterns impact exposure. However, this type of measurement error is likely to be non-differential [37], since this kind of mobility pattern may have been controlled by self-matching (the same week day for a month) in the time-stratified case-crossover design.

No standard definition exists for heat waves, either in the scientific literature or in policy, but heat waves are typically defined based on intensity, referring to level of temperature, and duration, i.e., consecutive days of high temperature. We analyzed five definitions with varying thresholds of temperature intensity, and a duration threshold of two or more days. Future studies could consider the impacts of alternative heat wave definitions with higher duration thresholds, as well the timing of heat waves within seasons, to examine whether, for example, the impacts of the first heat wave of the hot season are different from those of later ones. Additional work is also needed to examine the effect modification of certain environmental factors, such as humidity and air pollution, and individual- and contextual-level socioeconomic characteristics.

Despite our research limitations, few studies of heat waves have been based on individual-level temperature estimates, allowing exposure estimates for a given day to vary within the same city, and, to our knowledge, this is the first such study carried out in Brazil. Individual-level estimates of exposure are an interesting approach to estimating thermal exposure in large cities since temperature levels can be heterogeneous within cities, mainly because of the occurrence of heat islands. Future studies should explore more robust methods for estimating individual-level meteorological exposure. Another important study contribution is that our findings suggest that women, older people, and particularly older women, constitute the groups most vulnerable to the effects of heat waves, which may support the design and implementation of warning systems and heat-health action plans, enabling health and related sectors to adapt to climate change.

## Supporting information

**S1 Fig. Location of monitoring stations for individual-level exposures to temperature (T) and humidity (H) using the inverse distance weighted method, in the city of Rio de Janeiro, Brazil.** The heat map of deaths was created using a 500 m search radius. Source of the cartographic database: Brazilian Institute of Geography and Statistics.
(TIF)

**S2 Fig. Directed acyclic graph (DAG) with our assumptions regarding the causal effect of heat waves on cardiovascular and respiratory mortality.** Causal pathways between exposure and outcome are represented by green arrows; noncausal pathways (back-door paths) by red arrows; unmeasured factors by grey nodes; confounders by red nodes; and factors controlled by design by white nodes. The sufficient adjustment set of variables to estimate the total effect of heat waves on mortality included humidity and public holidays.
(TIF)

**S3 Fig. Lag effects of heat waves on cardiovascular and respiratory mortality (comparing heat wave to non-heat wave days) over 5 days in Rio de Janeiro, Brazil.** We used a natural cubic spline with 4 degrees of freedom for the lag effect. Effect estimates are reported as odds ratios (OR), and the dashes represent 95% confidence intervals.
(PDF)

**S1 Table. Percentage of missing data imputed in the daily series of meteorological variables for each monitor.**
(DOCX)

**S2 Table. Root mean square error (RMSE) and mean absolute error (MAE) values of cross-validation for the different sets of inverse distance weighting parameters.**
(DOCX)

**S3 Table. Mean threshold values for each heat wave definition, based on individual threshold values.** For all definitions, we used a duration of 2 days or more.
(DOCX)

**S4 Table. Results of sensitivity analysis.**
(DOCX)

## Author Contributions

**Conceptualization:** Ismael H. Silveira, Taísa Rodrigues Cortes, Michelle L. Bell, Washington Leite Junger.

**Data curation:** Ismael H. Silveira, Taísa Rodrigues Cortes, Washington Leite Junger.

**Formal analysis:** Ismael H. Silveira, Washington Leite Junger.

**Funding acquisition:** Michelle L. Bell.

**Investigation:** Ismael H. Silveira, Taísa Rodrigues Cortes, Michelle L. Bell.

**Methodology:** Ismael H. Silveira, Taísa Rodrigues Cortes, Washington Leite Junger.

**Project administration:** Michelle L. Bell, Washington Leite Junger.

**Resources:** Michelle L. Bell, Washington Leite Junger.

**Software:** Ismael H. Silveira, Taísa Rodrigues Cortes, Washington Leite Junger.

**Supervision:** Washington Leite Junger.

**Validation:** Taísa Rodrigues Cortes.

**Writing – original draft:** Ismael H. Silveira, Taísa Rodrigues Cortes.

**Writing – review & editing:** Michelle L. Bell, Washington Leite Junger.

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
