## [Decision Letter · Decision Letter 0]

4 Jan 2023

PONE-D-22-33396Effects of heat waves on cardiovascular and respiratory mortality in Rio de Janeiro, BrazilPLOS ONE

Dear Dr. Silveira,

Thank you for submitting your manuscript to PLOS ONE. After careful consideration, we feel that it has merit but does not fully meet PLOS ONE’s publication criteria as it currently stands. Therefore, we invite you to submit a revised version of the manuscript that addresses the points raised during the review process.

We look forward to receiving your revised manuscript.

Kind regards,

Nir Y. Krakauer

Academic Editor

PLOS ONE

3. We note that S1 Figure in your submission contain [map/satellite] images which may be copyrighted. All PLOS content is published under the Creative Commons Attribution License (CC BY 4.0), which means that the manuscript, images, and Supporting Information files will be freely available online, and any third party is permitted to access, download, copy, distribute, and use these materials in any way, even commercially, with proper attribution. For these reasons, we cannot publish previously copyrighted maps or satellite images created using proprietary data, such as Google software (Google Maps, Street View, and Earth). For more information, see our copyright guidelines: http://journals.plos.org/plosone/s/licenses-and-copyright.

a. You may seek permission from the original copyright holder of S1 Figure to publish the content specifically under the CC BY 4.0 license. 

Reviewers' comments:

Reviewer's Responses to Questions

**Comments to the Author**

1. Is the manuscript technically sound, and do the data support the conclusions?

Reviewer #1: Yes

Reviewer #2: No

2. Has the statistical analysis been performed appropriately and rigorously? 

Reviewer #1: Yes

Reviewer #2: No

3. Have the authors made all data underlying the findings in their manuscript fully available?

Reviewer #1: No

Reviewer #2: No

4. Is the manuscript presented in an intelligible fashion and written in standard English?

Reviewer #1: Yes

Reviewer #2: Yes

5. Review Comments to the Author

Reviewer #1: I read with interest this paper evaluating the effects of heat waves on cause-specific mortality in Rio de Janeiro.

The paper is well-written and well-structured. The statistical methods (conditional logistic regression) are coherent with a time-stratified case cross-over design. The results are coherent with those published previously in the literature and followed by an adequate discussion.

I have only some reservation about three aspects:

1) usually aboslute humidity is highly correlated with ambient temperature, so I guess could be also highly correlated with heat waves indicator. I wonder if could be more robust presenting as main results the estimates not adjusted by absolute huidity and in the sensitivity analysis presents the results adjusted by absolute humidity. Related to this point it would be interesting to explore the possible modifier effect of humidity instead of the confounding effect.

2) Looking at the lagged effect on respiratory deaths it looks that a longer lag should be considered for this cause. Does the lag curve tend to the null effect when considering 10 days of lags?

3) I would have considered in the main analysis only warmest months (November to March)

Reviewer #2: Overall, I think the manuscript would be quite interesting to both the general public and health and climate scientists, however, there are areas lacking detail which need to be addressed, in particular regarding the meteorological aspects. The manuscript is not always clear, it is silent on several methodological aspects/procedures. Also, there are other aspects that, in my opinion, can mislead the reader, especially regarding meteorological data/analysis. In general, I was not able to analyze properly the results and discussion sections since I have several doubts regarding the data and methods used.

Overall, I suggest that the authors consider all the major and minor changes suggested below which I hope will help the authors to improve their manuscript.

Introduction

The first sentence of the introduction states that Heat waves, meteorological events characterized by high temperatures sustained for two or more consecutive days. Defining heat wave as a sequence of high temperatures does not correspond to any of the standard HW definitions. I suggest: HW are meteorological events characterized by UNUSUAL high temperatures. Otherwise, the inattentive reader might think that all high temperatures correspond to heat waves. Moreover, along the manuscript, authors say that there is no universal definition for a heat wave (L138, L292). Accordingly, I suggest a more general definition, such as: HW are meteorological events characterized by UNUSUAL high temperatures, sustained for PROLONGED periods. Finally, the link for the WHO site is not working.

L65 – Reference [9] please refer to the lasted IPCC report.

L76-83. Please consider taking into account a recent national-level review: https://nyaspubs.onlinelibrary.wiley.com/doi/abs/10.1111/nyas.14887; Also, I suggest to consider other related works for Rio de Janeiro: https://doi.org/10.1016/j.scitotenv.2018.09.060

L84. Please define the study period here.

Methods

I feel some parts in the data and methods description are not well explained. The methods, as such, do not clearly describe how the research was conducted.

L94-95. I suppose that mortality data belongs from Tabnet system http://tabnet.rio.rj.gov.br/. Please clarify in the text.

Mortality data: why using just the period from 2012 to 2017? Why not using the entire available data? Please clarify in the text.

Weathar data.

First it is important to describe the name and code of each meteorological station and the website used to download the dataset. Second, it is important to clarify the period of the dataset and if all stations provide data for the same period. Third, is it essential that the author clarify how they calculate daily mean temperature (average value of all hourly data; average between Tmax and Tmin?). As far as I know, both the source of the data used here, INMET and ICEA, provide daily mean temperature and relative humidity already calculated according to the rules provided by the World Meteorological Organization. Accordingly, authors should clarify why not using a pre-calculated information. Moreover, I fail to understand how and why you need to fill the gaps of missing values. Why not just ignoring them? If you have only selected station with less than 20% missing data per year, then missing values will not compromise your analysis. L114-116, very difficult to follow. What do you mean here with modeled temporal components? Why? It is also difficult to understand table S1. Please clarify if TABLE S1 is about hourly or daily missing values for the entire period.

HW definition

A major concern of mine is about the definition and identification of the periods under HW. Mean temperature is not the best option to define HW events. A recent paper (https://doi.org/10.1007/s00484-020-01908-x) analyzed the relationship between various HW indices and mortality in Rio de Janeiro for the 2000 to 2015 period and concluded that the EHF index showed a better predictive capacity than Tmax and Tmean. EHF has the advantage of considering both the maximum and the minimum temperatures and to consider the acclimatization of the population during an HW preceding period of 30 days.

Moreover, regardless of the use of Tmean, the definition of HW as based of a fixed threshold for the entire period is not appropriate (as in table S3). Authors should be aware about the importance of accommodating the strong seasonal cycle of temperature and humidity variables. It is also important to clarify the climatological period used. For instance, looking at Table S3, by using a threshold of 29 °C during summer you will detect HW in almost all days. But during the winter, the situation could be very different. It would be nice to see a figure showing the time series of Tmean and the identification of periods in HW conditions. I think it would be important if you could make a clear statement about this. Otherwise, the inattentive reader might think that heat waves occur every day in the summer. In general, a different percentile threshold value is computed for each day of the year in order to take into account the seasonal cycle. This is also important because you say in L185-186 that you restricted your analysis to the “hot season” (November to March).

I also failed to understand table S3 the HW definition in the spatial context. You have 13 stations, why just showing one value in table S3? Each station should have its own thresholds.

Finally, authors say that duration is less important than intensity. Could you please me more assertive in this attribution? Also, please explain how you have defined intensity.

Please explain how humidity data was used in your study and why it is important.

Study design and statistical analysis

This section is very hard to follow. It would be nice to have an explanation about the results expected by applying each procedure. Please clarify if the analysis was carried out for each weather station or not.

L185-186 you need to justify this choice. Please consider this paper that shows the greatest short-time (daily-scale) mortality peaks in Rio de Janeiro are observed during summer periods, https://doi.org/10.1007/s00484-020-01908-x

Results

Table 1. Again, I cannot understand the meaning of the values regarding weather variables here. This is an average for the 13 stations? For what period and for what time scale (hour, days, months, year)?

What do you mean by individual-level?

6. PLOS authors have the option to publish the peer review history of their article (what does this mean?). If published, this will include your full peer review and any attached files.

Reviewer #1: **Yes: **Francesco Sera

Reviewer #2: No

---

## [Author Response · Author response to Decision Letter 0]

2 Mar 2023

Reviewer #1:

I read with interest this paper evaluating the effects of heat waves on cause-specific mortality in Rio de Janeiro.

The paper is well-written and well-structured. The statistical methods (conditional logistic regression) are coherent with a time-stratified case cross-over design. The results are coherent with those published previously in the literature and followed by an adequate discussion.

I have only some reservation about three aspects:

1) usually aboslute humidity is highly correlated with ambient temperature, so I guess could be also highly correlated with heat waves indicator. I wonder if could be more robust presenting as main results the estimates not adjusted by absolute huidity and in the sensitivity analysis presents the results adjusted by absolute humidity. Related to this point it would be interesting to explore the possible modifier effect of humidity instead of the confounding effect.

Response: We chose to use absolute humidity because, according to Davis et al. (2016), relative humidity is highly correlated with ambient temperature and, in assessments of heat effect, a water-vapor mass-based variable, such as absolute humidity, is more appropriate and more frequently recommended than relative humidity. In the first version of our manuscript, we justified this choice, but we have moved this sentence to the statistical analysis section in order to clarify our reasons for this option (page 8, lines 195-196). In line with our causal model (S2 Fig.), humidity was considered as a confounder and was included in the analyzes as an adjustment variable, so we kept it in the model. The model without humidity was analyzed in our sensitivity analysis (S4 Table). However, we have added a recommendation to the discussion section for future studies to further explore the role of certain social and environmental factors, including humidity, in modifying the effects of heat waves on health (page 15, lines 338-340).

2) Looking at the lagged effect on respiratory deaths it looks that a longer lag should be considered for this cause. Does the lag curve tend to the null effect when considering 10 days of lags?

Response: The reviewer is correct regarding the greater effect of a 10-day lag period for respiratory mortality. We have substituted the lag period from 5 to 10 days in order to estimate the effects of heat waves on respiratory diseases and their respective subgroups. Adjustments have been made to the abstract (page 2, line 33), the methods section (page 8, lines 186-188), the results report (Table 1), and the sensitivity analysis (page 9, line 199).

3) I would have considered in the main analysis only warmest months (November to March)

Response: We accepted the reviewer's suggestion and reanalyzed our data restricted to warmer months (November to March). We have changed the methods and results (page 2, lines 24 and 30; page 7, line 162; page 9, line 209 and 2014; Table 1; Table 2; S3 Table; S4 Table).

Reviewer #2: 

Overall, I think the manuscript would be quite interesting to both the general public and health and climate scientists, however, there are areas lacking detail which need to be addressed, in particular regarding the meteorological aspects. The manuscript is not always clear, it is silent on several methodological aspects/procedures. Also, there are other aspects that, in my opinion, can mislead the reader, especially regarding meteorological data/analysis. In general, I was not able to analyze properly the results and discussion sections since I have several doubts regarding the data and methods used.

Overall, I suggest that the authors consider all the major and minor changes suggested below which I hope will help the authors to improve their manuscript.

Introduction

The first sentence of the introduction states that Heat waves, meteorological events characterized by high temperatures sustained for two or more consecutive days. Defining heat wave as a sequence of high temperatures does not correspond to any of the standard HW definitions. I suggest: HW are meteorological events characterized by UNUSUAL high temperatures. Otherwise, the inattentive reader might think that all high temperatures correspond to heat waves. Moreover, along the manuscript, authors say that there is no universal definition for a heat wave (L138, L292). Accordingly, I suggest a more general definition, such as: HW are meteorological events characterized by UNUSUAL high temperatures, sustained for PROLONGED periods. Finally, the link for the WHO site is not working.

Response: We have changed the text as suggested (page 2, lines 55-56) and corrected the link to the WHO website (page 16, line 355).

L65 – Reference [9] please refer to the lasted IPCC report.

Response: We have referred to the latest IPCC report as recommended (page 17, lines 380-384).

L76-83. Please consider taking into account a recent national-level review: https://nyaspubs.onlinelibrary.wiley.com/doi/abs/10.1111/nyas.14887; Also, I suggest to consider other related works for Rio de Janeiro: https://doi.org/10.1016/j.scitotenv.2018.09.060

Response: We have included both suggested references in the text (page 3, line 82; page 4, lines 88-90), as well as a recently published paper by our research team (Silveira et al. 2023) (page 4, line 85). The latter has also been included in the discussion (page 13, line 285). 

L84. Please define the study period here.

Response: We have included the study period as suggested (page 4, line 92).

Methods

I feel some parts in the data and methods description are not well explained. The methods, as such, do not clearly describe how the research was conducted.

L94-95. I suppose that mortality data belongs from Tabnet system http://tabnet.rio.rj.gov.br/. Please clarify in the text.

Response: As described in the manuscript, mortality data were obtained from Rio de Janeiro’s Municipal Health Department, rather than from the Tabnet system. We used this data source because we needed residential addresses to geocode the deaths in order to obtain the individual-level exposure estimates. Data from Tabnet are publicly available but do not provide residential addresses.

Mortality data: why using just the period from 2012 to 2017? Why not using the entire available data? Please clarify in the text.

Response: During the project, we geocoded the mortality data from 2012 onwards, stopping in 2017. The geocoding of a large volume of health data requires considerable effort, which is why we used a 5-year period, as described in a previous article (Cortes et al. 2021). It is our understanding that 5 years is long enough to estimate the heatwave mortality effect in a large population.

Weathar data.

First it is important to describe the name and code of each meteorological station and the website used to download the dataset. Second, it is important to clarify the period of the dataset and if all stations provide data for the same period. Third, is it essential that the author clarify how they calculate daily mean temperature (average value of all hourly data; average between Tmax and Tmin?). As far as I know, both the source of the data used here, INMET and ICEA, provide daily mean temperature and relative humidity already calculated according to the rules provided by the World Meteorological Organization. Accordingly, authors should clarify why not using a pre-calculated information. Moreover, I fail to understand how and why you need to fill the gaps of missing values. Why not just ignoring them? If you have only selected station with less than 20% missing data per year, then missing values will not compromise your analysis. L114-116, very difficult to follow. What do you mean here with modeled temporal components? Why? It is also difficult to understand table S1. Please clarify if TABLE S1 is about hourly or daily missing values for the entire period.

Response: 1) The details (code and source) of each meteorological station are provided in the S1 Table. We have added, in the of S1 Table footnote and in the manuscript (page 5, line 118), additional information about weather data availability containing the websites for download. 2) We have clarified that meteorological data were obtained for the study period (page 5, line 112). 3) Some meteorological stations provided daily measures of temperature and humidity (including minimum, mean and maximum values), while others provided hourly values (S1 Table) for which daily means were calculated by averaging these hourly values. We have included this information in the main text (page 5, lines 122-124) and details about the frequency of each station dataset in the S1 Table. 4) To the best of our knowledge, missing data are an important concern in epidemiological studies. Even using stations with less than 20% missing data per year, it seems advantageous to us to impute the missing values. In the imputation method we used (Junger and Ponce de Leon 2015), the components of a time series (long term trend and seasonality) need to be modeled using certain parameters, as described. Additional details about this method can be found in the reference article (Junger and Ponce de Leon 2015). 5) We have included additional details regarding the frequency of crude weather data in the S1 Table.

HW definition

A major concern of mine is about the definition and identification of the periods under HW. Mean temperature is not the best option to define HW events. A recent paper (https://doi.org/10.1007/s00484-020-01908-x) analyzed the relationship between various HW indices and mortality in Rio de Janeiro for the 2000 to 2015 period and concluded that the EHF index showed a better predictive capacity than Tmax and Tmean. EHF has the advantage of considering both the maximum and the minimum temperatures and to consider the acclimatization of the population during an HW preceding period of 30 days.

Response: As noted in the manuscript, there is no single universal definition of a heat wave. According to a broad systematic review (Xu et al. 2016), different temperature indicators, including mean temperature have been used to define heat wave events in the literature. 

The authors even report that previous studies found that mean temperature was a better predictor of mortality, since it is more likely to represent the heat level over 24 hours. Geirinhas et al. (2020) concluded that EHF had a better predictive capacity than Tmean, based on reduced dispersion around the curve in high EHF values and corresponding mortality data, and the lower RMSE obtained for EHF. But it is important to note that they used a different regression model from ours, and in addition, Tmean had a similar relationship with mortality levels, and quite a similar RMSE value. In this way, we believe that there is no evidence that Tmean is not an appropriate predictor of HW-related mortality risk.

Moreover, regardless of the use of Tmean, the definition of HW as based of a fixed threshold for the entire period is not appropriate (as in table S3). Authors should be aware about the importance of accommodating the strong seasonal cycle of temperature and humidity variables. It is also important to clarify the climatological period used. For instance, looking at Table S3, by using a threshold of 29 °C during summer you will detect HW in almost all days. But during the winter, the situation could be very different. It would be nice to see a figure showing the time series of Tmean and the identification of periods in HW conditions. I think it would be important if you could make a clear statement about this. Otherwise, the inattentive reader might think that heat waves occur every day in the summer. In general, a different percentile threshold value is computed for each day of the year in order to take into account the seasonal cycle. This is also important because you say in L185-186 that you restricted your analysis to the “hot season” (November to March).

Response: Since there is no universal definition for heat waves, some researches and meteorological organizations define these using a fixed or variable threshold to define heat wave intensity. The former can be based on a relative threshold (i.e., some percentile of a temperature distribution, which may refer to the study period or to a preceding climatological period) or absolute threshold (according to some physiological evidence) (Zuo et al. 2015; Xu et al. 2016; Guo et al. 2017). In our study, we used a fixed temperature threshold based on temperature distribution over the study period, but now we restricted our data to the hot season. In addition, it is important to note that each participant has their own temperature distribution, estimated according their residential address using the IDW method. 

The S3 Table shows the mean values of the thresholds based on the average individual values. We restricted our analysis to the warmer months (November to March) (page 7, lines 153, 162-163). We didn’t include a figure with the Tmean times series because each participant has their own exposure estimation. 

I also failed to understand table S3 the HW definition in the spatial context. You have 13 stations, why just showing one value in table S3? Each station should have its own thresholds.

Response: We estimated a temperature exposure series for each individual, based on their respective residential address, and then classified the days in the series as heat wave days or not. The values presented in the S3 Table are the average values of the individual thresholds. We have clarified this in the text (page 5, lines 120-121; and in the S3 Table title). 

Finally, authors say that duration is less important than intensity. Could you please me more assertive in this attribution? Also, please explain how you have defined intensity.

Response: In fact, we only suggest that duration tends to be less important than intensity, based on observations from the cited systematic review. We have explained that the definition of intensity was based on 5 different percentiles from the individual temperature series. We have clarified this (page 7, lines 153-155).

Please explain how humidity data was used in your study and why it is important.

Response: In our analysis, we considered humidity to be a confounder, as presented in our causal model (S2 Fig; page 8, lines 177-179 and lines 192-196).

Study design and statistical analysis

This section is very hard to follow. It would be nice to have an explanation about the results expected by applying each procedure. Please clarify if the analysis was carried out for each weather station or not.

Response: We started this section by explaining our epidemiological study design, followed by our causal model (in order to explain the relationship between the variables used in our analysis), and our regression model and parameters, ending with our sensitivity analysis. We used weather station data to derive individual level exposure estimates, based on the IDW method, as described in the manuscript (page 2, lines 27-28; page 5, lines 130-148).

L185-186 you need to justify this choice. Please consider this paper that shows the greatest short-time (daily-scale) mortality peaks in Rio de Janeiro are observed during summer periods, https://doi.org/10.1007/s00484-020-01908-x

Response: We have already presented the reasons for adopting the parameters in the main analysis (page 8, lines 187-192). Lines 185-186 (page 9, lines 197-204 in the new version) refer to certain sensitivity analyses, which are commonly conducted in order to check whether the results remain robust after changing some of the analysis parameters. 

Results

Table 1. Again, I cannot understand the meaning of the values regarding weather variables here. This is an average for the 13 stations? For what period and for what time scale (hour, days, months, year)?

Response: Average temperature and humidity were based on weather stations and the mean of individual-level variables were based on the individual-level exposure estimation. We have included a foot note in Table 1 to clarify this (page 10, lines 216-217).

What do you mean by individual-level?

Response: In this study, we estimated the individual-level exposure to temperature and humidity. The individual-level variables refer to the exposure estimated in a buffer centered on the participant address based on the inverse distance weighted method. We report this in several places throughout the main text (page 2, line 27; page 4, lines 94-97; page 6, lines 129-146; page 12, line 257; page 14, lines 306-308; page 15, line 339).

References

Cortes TR, Silveira IH da, Junger WL. Improving geocoding matching rates of structured addresses in Rio de Janeiro, Brazil. Cad Saude Publica [Internet]. 2021;37(7). Available from: http://www.scielo.br/scielo.php?script=sci_arttext&pid=S0102-311X2021000706001&tlng=en

Davis RE, McGregor GR, Enfield KB. Humidity: A review and primer on atmospheric moisture and human health. Environ Res. 2016 Jan;144:106–16. 

Geirinhas JL, Russo A, Libonati R, Trigo RM, Castro LCO, Peres LF, et al. Heat-related mortality at the beginning of the twenty-first century in Rio de Janeiro, Brazil. Int J Biometeorol [Internet]. 2020 Aug 20;64(8):1319–32. Available from: http://link.springer.com/10.1007/s00484-020-01908-x

Guo Y, Gasparrini A, Armstrong BG, Tawatsupa B, Tobias A, Lavigne E, et al. Heat wave and mortality: A multicountry, multicommunity study. Environ Health Perspect. 2017;125(8). 

Junger WL, Ponce de Leon A. Imputation of missing data in time series for air pollutants. Atmos Environ [Internet]. 2015;102:96–104. Available from: http://dx.doi.org/10.1016/j.atmosenv.2014.11.049

Silveira IH, Hartwig SV, Moura MN, Cortes TR, Junger WL, Cirino G, et al. Heat waves and mortality in the Brazilian Amazon: Effect modification by heat wave characteristics, population subgroup, and cause of death. Int J Hyg Environ Health [Internet]. 2023 Mar;248:114109. Available from: https://linkinghub.elsevier.com/retrieve/pii/S1438463922001924

Xu Z, FitzGerald G, Guo Y, Jalaludin B, Tong S. Impact of heatwave on mortality under different heatwave definitions: A systematic review and meta-analysis. Environ Int [Internet]. 2016 Apr;89–90:193–203. Available from: https://linkinghub.elsevier.com/retrieve/pii/S0160412016300411

Zuo J, Pullen S, Palmer J, Bennetts H, Chileshe N, Ma T. Impacts of heat waves and corresponding measures: a review. J Clean Prod [Internet]. 2015 Apr;92:1–12. Available from: https://linkinghub.elsevier.com/retrieve/pii/S0959652614013754

---

## [Decision Letter · Decision Letter 1]

20 Mar 2023

Effects of heat waves on cardiovascular and respiratory mortality in Rio de Janeiro, Brazil

PONE-D-22-33396R1

Dear Dr. Silveira,

We’re pleased to inform you that your manuscript has been judged scientifically suitable for publication and will be formally accepted for publication once it meets all outstanding technical requirements.

Kind regards,

Nir Y. Krakauer

Academic Editor

PLOS ONE

Reviewers' comments:

Reviewer's Responses to Questions

**Comments to the Author**

1. If the authors have adequately addressed your comments raised in a previous round of review and you feel that this manuscript is now acceptable for publication, you may indicate that here to bypass the “Comments to the Author” section, enter your conflict of interest statement in the “Confidential to Editor” section, and submit your "Accept" recommendation.

Reviewer #1: All comments have been addressed

2. Is the manuscript technically sound, and do the data support the conclusions?

Reviewer #1: Yes

3. Has the statistical analysis been performed appropriately and rigorously? 

Reviewer #1: Yes

4. Have the authors made all data underlying the findings in their manuscript fully available?

Reviewer #1: No

5. Is the manuscript presented in an intelligible fashion and written in standard English?

Reviewer #1: Yes

6. Review Comments to the Author

Reviewer #1: The authors answered positevely to all my comments.

I think the manuscript has improved from the origianl submission and could give a contribution on the litterature on health effects of heat waves.

7. PLOS authors have the option to publish the peer review history of their article (what does this mean?). If published, this will include your full peer review and any attached files.

Reviewer #1: **Yes: **Francesco Sera

---

## [Editor Report · Acceptance letter]

22 Mar 2023

PONE-D-22-33396R1 

Effects of heat waves on cardiovascular and respiratory mortality in Rio de Janeiro, Brazil 

Dear Dr. Silveira:

I'm pleased to inform you that your manuscript has been deemed suitable for publication in PLOS ONE. Congratulations! Your manuscript is now with our production department. 

Kind regards, 

on behalf of

Dr. Nir Y. Krakauer 

Academic Editor

PLOS ONE